# Metabolic Signature of Warburg Effect in Cancer: An Effective and Obligatory Interplay between Nutrient Transporters and Catabolic/Anabolic Pathways to Promote Tumor Growth

**DOI:** 10.3390/cancers16030504

**Published:** 2024-01-24

**Authors:** Marilyn Mathew, Nhi T. Nguyen, Yangzom D. Bhutia, Sathish Sivaprakasam, Vadivel Ganapathy

**Affiliations:** Department of Cell Biology and Biochemistry, Texas Tech University Health Sciences Center, Lubbock, TX 79430, USA; marilyn.mathew@ttuhsc.edu (M.M.); nhi.t.nguyen@ttuhsc.edu (N.T.N.); yangzom.d.bhutia@ttuhsc.edu (Y.D.B.); sathish.sivaprakasam@ttuhsc.edu (S.S.)

**Keywords:** oncogenes, aerobic glycolysis, lactate receptors, nutrient transporters, glutamine addiction, one-carbon metabolism, glutaminolysis, reductive carboxylation, oncometabolites, tumor microenvironment

## Abstract

**Simple Summary:**

Cancer represents unrestricted growth with the removal of conventional brakes that control growth under normal conditions. This requires novel mechanisms to provide metabolic energy to fuel the rapid growth and also macromolecules to support cell renewal. This unique need in cancer cells is accomplished by an efficient interplay between selective nutrient transporters and the reprogramming of cellular metabolism that modifies specific catabolic and anabolic pathways. These modified biochemical pathways generate certain metabolites that are seen at high levels only in cancer cells, and reroute signaling cascades, alter gene expression profiles, and exert biological effects in support of the growth and proliferation of cancer cells. A clear understanding of the metabolic signature that is unique to cancer cells is necessary not only to appreciate how the unrestricted growth is accomplished in cancer but also to exploit these cancer-cell-specific nutrient transporters and metabolic pathways as drug targets to develop new anticancer therapeutics.

**Abstract:**

Aerobic glycolysis in cancer cells, originally observed by Warburg 100 years ago, which involves the production of lactate as the end product of glucose breakdown even in the presence of adequate oxygen, is the foundation for the current interest in the cancer-cell-specific reprograming of metabolic pathways. The renewed interest in cancer cell metabolism has now gone well beyond the original Warburg effect related to glycolysis to other metabolic pathways that include amino acid metabolism, one-carbon metabolism, the pentose phosphate pathway, nucleotide synthesis, antioxidant machinery, etc. Since glucose and amino acids constitute the primary nutrients that fuel the altered metabolic pathways in cancer cells, the transporters that mediate the transfer of these nutrients and their metabolites not only across the plasma membrane but also across the mitochondrial and lysosomal membranes have become an integral component of the expansion of the Warburg effect. In this review, we focus on the interplay between these transporters and metabolic pathways that facilitates metabolic reprogramming, which has become a hallmark of cancer cells. The beneficial outcome of this recent understanding of the unique metabolic signature surrounding the Warburg effect is the identification of novel drug targets for the development of a new generation of therapeutics to treat cancer.

## 1. Introduction

About 100 years ago, Otto Warburg made an interesting observation: cancer cells when cultured in vitro under normal oxygen levels (i.e., 20% or 160 mmHg) as in physiological conditions or tumors growing in the body in vivo consume glucose much more than normal cells but convert this glucose predominantly into lactic acid, which is released into the culture medium [1,2]. This was unexpected because lactic acid is the end product of glucose breakdown only under conditions of oxygen deficit (i.e., hypoxia), a pathway known as “anerobic glycolysis”. In contrast to this normal process, cancer cells convert glucose into lactic acid in the presence of oxygen, thus leading to the coining of the term “aerobic glycolysis” to describe the observation made by Warburg in cancer cells. Though interesting and unexpected, the importance of this observation was not recognized for several decades. A part of the reason for this was the discovery of oncogenes and tumor suppressor genes and the domination of the idea in the field of cancer biology that the protein products of these genes are the principal drivers of cancer growth. Surprisingly, however, investigations into the molecular targets of these oncogenes and tumor suppressor genes began to underscore the importance of metabolic pathways as the likely mediators of these genes in cancer growth. As a result, the interest in the field of cancer biology has shifted in the past couple of decades to cancer-cell-specific metabolism. Naturally, the starting point for this shift was the Warburg effect, which represents the first metabolic pathway to be discovered that is specific to cancer cells. What followed in subsequent years in this field is simply a logical extension of the original observation by Warburg.

The greater than normal consumption of glucose in cancer cells brought attention to glucose transporters, which deliver glucose into these cells. Since lactic acid is generated at high levels, cancer cells must find ways to prevent cellular acidification and get rid of lactic acid. This shifted the focus to lactate transporters that mediate the transfer of lactate and H^+^ across the plasma membrane [3,4]. Then came the discovery that lactate controls the proteasomal degradation of hypoxia-inducible factor-1α (HIF-1α), thus increasing HIF-1α protein levels in cancer cells even in the presence of normal oxygen, a condition now described as “pseudohypoxia” [5]. Since cancer cells release massive amounts of lactic acid into the tumor microenvironment, which results in extracellular acidification, investigations began to explore the role of acidic pH in the tumor microenvironment in cancer growth. This led to the focus on H^+^-coupled transporters for various nutrients such as amino acids, peptides, citrate, folate, and iron, which might use the extracellular acidic pH to effectively transfer these important nutrients to cancer cells from the extracellular medium [6]. With the increased rate of glycolysis came increased levels of metabolic intermediates in the pathway. This brought attention to the use of these intermediates for the anabolic pathways to generate amino acids such as serine and glycine, which are obligatory as the source of one-carbon moieties for one-carbon metabolism involved in the synthesis of purines, pyrimidines, and thymidine monophosphate (TMP) [7,8].

Cancer cells must have a greater need than normal cells for metabolic energy to support their high rate of proliferation, but the glucose–lactic acid pathway that occurs in these cells generates only a fraction of energy compared to the complete oxidation of glucose to CO_2_ (2 ATP versus 32 ATP). This raised the possibility of other metabolic pathways generating ATP, which led to the discovery of the obligate dependence of cancer cells on extracellular glutamine (glutamine addiction). Subsequent research showed that glutamine is used in cancer cells not only for ATP generation but also for lactic acid production (glutaminolysis) and fatty acid synthesis (reductive carboxylation) [9,10,11]. This also brought attention to amino acid transporters in the plasma membrane of cancer cells, which deliver not only glutamine to satisfy the “glutamine addiction” but also serine and glycine to fuel the one-carbon metabolism in addition to serine and glycine that are synthesized from the glycolytic intermediates [12,13,14,15]. Cancer cells obtain amino acids not only from the extracellular medium but also from the lysosomal degradation of intracellular proteins via autophagy and extracellular proteins via macropinocytosis [16,17]. This necessitated studies on amino acid transporters in the lysosomal membrane in cancer cells that transfer amino acids from lysosomes into cytoplasm for subsequent use in metabolic pathways [18,19]. This also integrated amino acid nutrition to mTORC1 signaling because of the close association of mTORC1-associated proteins with the lysosomal membrane, a critical signaling pathway necessary for cancer cell proliferation and growth [18,19].

Cancer cells are also addicted to iron and heme because of their essential role in a multitude of metabolic processes involved in energy production as well as catabolic and anabolic processes [20,21]. Some of the critical steps in heme synthesis occur within the mitochondrial matrix, including the first regulatory step in the pathway, which combines glycine with succinyl-CoA to generate δ-aminolevulinate. In addition, one-carbon metabolism also participates in important biochemical processes within the mitochondrial matrix that require serine. This brings attention to transporters in the mitochondrial membrane that deliver serine and glycine from cytoplasm to the mitochondrial matrix [22]. Furthermore, with the accumulation of excess iron in cancer cells comes the risk of oxidative stress and the iron-dependent cell death process (ferroptosis) [23,24,25]. Therefore, cancer cells must enhance their antioxidant machinery to protect themselves from these detrimental processes. This led to the focus on the glutathione/glutathione peroxidase system and pentose phosphate pathway, which produce the reducing equivalent NADPH necessary for the antioxidant machinery [26]. In addition, since cysteine is the rate-limiting amino acid for glutathione synthesis, studies began on amino acid transporters in the plasma membrane and lysosomal membrane that provide this critical amino acid to cancer cells to support the enhanced glutathione production [27,28,29].

The original explanation for “aerobic glycolysis” thought by Warburg was that mitochondria are damaged in cancer cells and therefore the oxygen-dependent metabolism of pyruvate in mitochondria is impaired. This means that almost all the metabolic energy needed for the cancer cells comes from the conversion of glucose to lactic acid. This view has now been revised considerably based on glutamine metabolism within the mitochondria in cancer cells. The source of energy production simply shifts to a significant extent from glycolysis to glutaminolysis with intact mitochondrial function necessary for the latter process. To support their growth and proliferation, cancer cells use the intermediates in the enhanced glycolytic pathway to feed into other metabolic pathways instead for energy production. 

What follows in this review is a detailed description of these various components of cancer-cell-specific metabolic pathways and the transporters related to them. Since these pathways and the transporters have been reprogrammed in cancer cells as a consequence and in conjunction with “aerobic glycolysis”, they represent the metabolic signature of the Warburg effect.

## 2. Hypoxia and Anaerobic Glycolysis in Normal Cells

Glucose can be metabolized completely into CO_2_ and H_2_O in cells with mitochondria when oxygen is available. This complete oxidation of glucose involves glycolysis (glucose → pyruvate) in cytoplasm and the pyruvate dehydrogenase (PDH) and citric acid cycle (pyruvate → CO_2_) in the mitochondrial matrix (Figure 1A). However, neither glycolysis nor the PDH/citric acid cycle involves oxygen. When glucose goes through glycolysis and the PDH/citric acid cycle, it generates reducing equivalents NADH and FADH_2_, which enter the electron transport chain, and oxidative phosphorylation, where oxygen is used to convert these reducing equivalents back to NAD^+^ and FAD with the concomitant production of ATP. The entire process results in the generation of 32 ATP per glucose. This “aerobic glycolysis” associated with the complete oxidation of glucose in normal cells is subject to negative feedback regulation by ATP, which inhibits phosphofructokinase-1 (PFK-1), the most important rate-limiting enzyme in glycolysis. In other words, the aerobic glycolysis in normal cells is self-limiting, controlled by the energy status of the cell.

When oxygen is deficient in normal cells, mitochondrial oxidation of pyruvate that is generated in glycolysis in the cytoplasm is impaired due to suppression of the electron transport chain (ETC) and oxidative phosphorylation (OXPHOS). Even though most of the NADH from glucose oxidation is produced within the mitochondria, the step mediated by glyceraldehyde-3-phosphate dehydrogenase (GAPDH) in glycolysis also generates NADH. If this NADH cannot be oxidized back to NAD^+^ due to defective ETC/OXPHOS as under hypoxic conditions, the reaction mediated by GAPDH cannot continue because of the lack of NAD^+^. This forces the conversion of pyruvate to lactate in the cytoplasm by lactate dehydrogenase (LDH), a reaction that substitutes for ETC/OXPHOS to convert NADH to NAD^+^ but without O_2_ consumption and ATP production (Figure 1B). Now, glycolysis can continue because of the functional coupling between GAPDH and LDH via NADH/NAD^+^ recycling, the entire process occurring in cytoplasm. This process where glucose gets converted to lactic acid in normal cells under hypoxic conditions is called “anaerobic glycolysis”. Interestingly, the same process occurs in erythrocytes even in the presence of oxygen because of the absence of mitochondria. Consequently, lactic acid is the end product of glycolysis in normal cells only under hypoxic conditions whereas erythrocytes generate lactic acid in glycolysis all the time.

## 3. Aerobic Glycolysis in Cancer Cells

### 3.1. Mechanisms used to Facilitate “Aerobic Glycolysis” in Cancer Cells

As mentioned above, glycolysis in normal cells in the presence of oxygen is self-limiting and pyruvate gets converted to CO_2_. The conversion of glucose to lactic acid occurs in normal cells only under hypoxic conditions. In contrast, cancer cells metabolize glucose predominantly into lactic acid even in the presence of oxygen. The functional coupling between the reactions mediated by GAPDH and LDH is necessary for this process. Why does pyruvate not get oxidized to CO_2_ in cancer cells when oxygen is available? This is mostly due to the defective transport of pyruvate from cytoplasm into the mitochondrial matrix because of the cancer-associated downregulation of the pyruvate-carrier components MPC1 and MPC2 in the inner mitochondrial membrane [30] and the decreased catalytic activity of PDH due to an increased expression of PDH kinase-1 (PDK-1) and PDH kinase-3 (PDK-3) and consequent increased phosphorylation of PDH [31]. An additional mechanism for the inactivation of PDH in cancer cells involves signaling pathways associated with EGFR activation and mutant K-Ras [32]. In this mechanism, the glycolytic enzyme phosphoglycerate kinase gets phosphorylated and, as a result, translocates into the mitochondrial matrix, where it acts as a protein kinase to phosphorylate PDK-1 and increases its catalytic activity. Thus, the enhanced activity of PDK-1 in cancer cells is the consequence of increased expression as well as post-translational modification. As a result, pyruvate is prevented from mitochondrial oxidation, thus getting diverted to lactate production in the cytoplasm (pyruvic acid + NADH + H^+^ → lactic acid + NAD^+^) (Figure 1C). This drives aerobic glycolysis in cancer cells by providing NAD^+^ for the reaction mediated by GAPDH.

Since aerobic glycolysis in cancer cells generates only 2 ATP per glucose instead of 32 ATP when glucose gets converted to CO_2_, it raises the question as to the energy status of the cancer cells. The unrestricted growth and proliferation of cancer cells cannot occur when the cells are energy-deficient. The energy status for supporting rapid growth is maintained in cancer cells by two mechanisms: one by accelerating “aerobic glycolysis” with an increased conversion of glucose to lactic acid and the other by generating energy from the metabolism of amino acids, primarily glutamine (Figure 1C). Even though glucose → lactic acid in cancer cells generates only two ATP per glucose, more ATP can be produced if the pathway is enhanced to metabolize more glucose. This metabolic switch together with glutamine-derived ATP maintains the energy status of the cancer cells to fuel their growth and proliferation.

Since glycolysis is subject to negative feedback regulation by ATP at the level of PFK-1, how can cancer cells enhance glycolysis and at the same time generate ATP at levels even higher than in normal cells? This is achieved by an increased production of fructose-2,6-bisphosphate, another regulatory molecule for PFK-1. While ATP is an inhibitor of PFK-1, fructose-2,6-bisphosphate is an activator that annuls the inhibition by ATP (Figure 1C). The cellular levels of this activator are controlled by the bifunctional enzyme phosphofructokinase-2/fructose-2,6-bisphosphatase (PFKFB). Two isoforms of this enzyme, namely PFKFB3 and PFKFB4, are upregulated in cancer cells, leading to increased levels of fructose-2,6-bisphosphate to rescue glycolysis from the negative feedback regulation by ATP [33,34]. The levels of fructose-2,6-bisphosphate in cancer cells are also regulated by another mechanism involving the protein TIGAR (TP53-induced glycolysis and apoptosis inhibitor) [35]. The expression of TIGAR is downregulated in p53-mutant tumors. TIGAR possesses the catalytic activity of fructose-2,6-bisphosphatase and hence has the ability to degrade fructose-2,6-bisphosphate. This results in a reciprocal relationship between the levels of TIGAR and frunctose-2,6-bisphosphate. Since the expression of TIGAR is decreased in p53-deficient tumors, fructose-2,6-bisphosphate levels go up to maintain “aerobic glycolysis” by keeping PFK-1 active even in the presence of ATP.

There are two structurally distinct genes coding for LDH: LDH-A and LDH-B. Since the holoenzyme is a tetramer, LDH exists in five different isoforms depending on the composition of the two gene products in the tetramer. LDH1 consists of all four subunits being LDH-B whereas LDH5 consists of all four subunits being LDH-A. LDH2, LDH3, and LDH4 consist of varying mixtures of both LDH-A and LDH-B. The relative affinities of LDH-A and LDH-B for lactate and pyruvate make LDH-A more amenable to facilitate the conversion of pyruvate to lactate and LDH-B more amenable to facilitate the conversion of lactate to pyruvate [36]. The expression of LDH-A is upregulated in all cancers whereas the expression of LDH-B is downregulated in most cancers. This shift in the relative amounts of the two isoforms facilitates “aerobic glycolysis” in cancer cells to convert pyruvate to lactate.

### 3.2. Aerobic Glycolysis in Non-Malignant Cells and Anerobic Glycolysis in Malignant Cells

Interestingly, aerobic glycolysis, where the conversion of glucose to lactic acid occurs even in the presence of adequate oxygen, is observed in certain cell types unrelated to cancer. This phenomenon may or may not be connected to a high cell proliferation rate. The first example in this category is the role of astrocytes in the brain and retina as providers of metabolic fuel to neurons in the form of lactate [37]. Glucose is metabolized in astrocytes primarily to produce lactic acid, which is then supplied to neurons as an energy substrate; this process is not related to an increased proliferation of astrocytes. Another example is immune cells (monocytes and lymphocytes) during the adaptive response [38]. In this case, the purpose of aerobic glycolysis may be to generate ATP at a rapid rate and also regulate certain specific signaling metabolites (e.g., lactate, succinate). This happens in activated lymphocytes, including those present in the context of a tumor to mount an immune response against the tumor cells, where aerobic glycolysis is associated with increased cell proliferation for the expansion of cytotoxic T cells. The same phenomenon also occurs in monocytes exposed to bacterial and fungal cell wall components. Another example is the endothelial cells involved in vessel sprouting, which opt for aerobic glycolysis with the generation of lactic acid [38]. Here, lactate is likely to function as a signaling molecule to aid in various cellular processes necessary for the construction of new blood vessels. It is, however, interesting to note that the “stalk” cells and the “tip” cells associated with vessel sprouting do exhibit certain features, such as increased motility and invasion, similar to cancer cells.

Hypoxia and the resultant anaerobic glycolysis do occur in cancer cells present in solid tumors. The rate of proliferation of cancer cells exceeds the rate of formation of tumor-associated blood vessels. As a result, heterogeneity exists among cancer cells in terms of oxygen availability. Cancer cells that are located far away from blood vessels are subjected to hypoxia, and therefore these cells shift their glucose metabolism to anaerobic glycolysis as normal cells do under similar hypoxic conditions. But, metabolic reprogramming that occurs in cancer cells due to oncogenes and tumor suppressors may force these cells to continue to generate lactic acid from glucose even when an adequate blood supply becomes available to them, thus shifting from anaerobic glycolysis to aerobic glycolysis with the continued generation of lactic acid. 

### 3.3. Lactic Acidosis and Tumor Microenvironment

Cancer cells produce massive amounts of lactic acid as a result of not only “aerobic glycolysis” but also acceleration of the pathway. Unless this lactic acid is removed from the cells, cellular pH will become acidic and interfere with normal biological and metabolic processes, thus being detrimental to the survival of cancer cells. To escape from such a consequence, cancer cells export lactic acid across the plasma membrane and release it into the extracellular medium. This results in severe lactic acidosis in the tumor microenvironment, increasing the levels of lactate as well as H^+^. Lactate levels in the tumor microenvironment have been shown to be as high as 40 mM compared to normal levels of 1.5–2.5 mM in circulation [39,40]. Releasing lactic acid from cancer cells into the extracellular medium is critical for supporting “aerobic glycolysis” since acidic pH in cytosol will inhibit the activities of glycolytic enzymes. Additionally, a slightly basic intracellular pH is required for the activity of LDH, facilitation of cell proliferation, and escape from apoptosis [39]. At the same time, acidic extracellular pH limits normal cell growth and survival. Cancer cells are more adaptive to the acidic environment than normal cells, so they can survive by adjusting their metabolism. The death of normal cells in the tumor microenvironment clears the space for the tumor to grow. Moreover, a lower pH in the tumor microenvironment provides a favorable environment for cancer cells to suppress the immune cells and degrade the extracellular matrix, which facilitates cancer cells to survive, invade, and metastasize.

### 3.4. Lactate and Pseudohypoxia

When oxygen availability is low in cells, the protein level of hypoxia-inducible factor-1α (HIF-1α), a transcription factor, increases to reprogram the transcriptional profile in such a way that the cells can adapt and try to survive under the hypoxic conditions. The underlying mechanism for the reciprocal changes in HIF-1α protein in response to oxygen levels involves the oxygen-dependent regulation of HIF-1α protein degradation via proteasomes. This process is controlled by prolyl hydroxylases (PHDs), which hydroxylate certain prolyl residues in HIF-1α as a prerequisite for proteasomal degradation. PHDs use molecular oxygen for the hydroxylation step with α-ketoglutarate and Fe^2+^ as additional cofactors [41]. Therefore, in the presence of oxygen, HIF-1α gets degraded. When the oxygen supply is low, the catalytic activity of PHDs is impaired, thus leading to a decreased prolyl hydroxylation of HIF-1α and consequent prevention of proteasomal degradation. As such, an increase in HIF-1 protein is a hallmark of hypoxia. Interestingly, one of the isoforms of PHDs, namely PHD2, is inhibitable by lactate via blockade of the binding of α-ketoglutarate to the enzyme [42]. This inhibition occurs in the presence of normal oxygen supply. Accordingly, cancer cells that accumulate lactate due to “aerobic glycolysis” have reduced PHD2 activity, reduced proteasomal degradation of HIF-1α, and hence increased levels of HIF-1α protein. Normally HIF-1α levels go up only under hypoxic conditions whereas, in cancer cells, HIF-1α levels go up under normoxic conditions. In other words, cancer cells behave as if they are under hypoxic conditions even in the presence of adequate oxygen supply. This situation is described as “pseudohypoxia” and it is the consequence of an increased production of lactate in cancer cells. As detailed subsequently in this review, the “pseudohypoxia” with the associated increase in HIF-1α is beneficial for the growth and proliferation of cancer cells since this transcription factor plays a critical role in the reprogramming of various metabolic pathways and in the expression of multiple nutrient transporters.

### 3.5. Oncogenic Transcription Factors HIF-1α and c-Myc and Their Relevance to “Aerobic Glycolysis”

HIFs are heterodimeric proteins with two subunits: an oxygen sensitivity α subunit and a constitutively expressed β subunit [43]. There are three paralogs of the HIF-1α subunit, including HIF-1α, HIF-2α, and HIF-3α, and two paralogs of the HIF-1β subunit (ARNT and ARNT2) in humans [44]. Among the three paralogs of the α subunit, HIF-1α is mostly expressed in normal tissues throughout the human body, whereas HIF-2α possesses a specific tissue expression pattern [45]. HIF-1α and HIF-2α are overexpressed in cancer cells, which supports tumor growth by upregulating genes participating in tumor invasion and angiogenesis. HIF-3α has been studied less than HIF-1α and HIF-2α. The biological functions of HIF-3α have not been completely elucidated.

HIF-1α comprises an inhibitory domain (ID) (residues 576–785) inserted in between two transactivation domains, TAD-N (residues 531–575) and TAD-C (residues 786–826) [46,47]. These transactivation domains bind to coactivators, including (CBP)/p300, SRC-1, and TIF-2, to induce HIF-1α mRNA expression [48]. HIF-1α has a short half-life, approximately about 5–10 min, and is rapidly degraded under normoxic conditions [49]. HIF-1α mRNA, protein, and their activity are tightly regulated by O_2_ [48]. In particular, the ubiquitin-mediated process regulating the degradation of HIF-1α with the involvement of the tumor suppressor protein pVHL (von Hippel–Lindau) is an O_2_-dependent process, whereas HIF-1β expression is independent of the presence of O_2_ [45]. Prolyl hydroxylase domain (PHD) enzymes use oxygen to hydroxylate conserved proline residues (Pro402 and Pro564) in the oxygen-dependent degradation domain (ODDD) region of the α subunit and acetyltransferase arrest defective 1 (ARD-33 1) enzyme acetylates lysine (Lys532) under normoxic conditions [50]. VHL, a component of E3 ubiquitin, selectively binds to the hydroxylated HIF1α at TAD-N and subsequently induces HIF1α degradation [51]. In addition to this proteasomal degradation process that controls the protein levels of HIF-1α, a different mechanism participates in regulating the transcriptional activity of HIF-1α. A factor inhibiting HIF (FIH) is a Fe^2+^-dependent enzyme that interacts with ID and hydroxylates HIF-1α via residue asparagine (Asn803) at TAD-C [48]; this hydroxylation interferes with the transcriptional activity of HIF-1α. Together, VHL and FIH prevent coactivators from binding to HIF-1α transactivation domains as well as destabilize HIF-1α.

HIF-1α levels are elevated in cancer cells even under normoxic conditions because of the inhibition of PHD2 by lactate (pseudohypoxia). The resultant increase in the transcriptional activity of HIF-1α fuels aerobic glycolysis in cancer cells. The targets for HIF-1α include LDH-A, PFKFB3, PDK-1, and PDK-3; therefore, the expression and activities of these proteins are increased in cancer cells. LDH-A favors the conversion of pyruvate to lactate; PFKFB3 increases the levels of fructose-2,6-bisphosphate, which then prevents the ATP-dependent inhibition of PFK-1; PDK-1 and PDK-3 inactivate PDH and hence interfere with the mitochondrial oxidation of pyruvate.

The Myc gene family is well known to play roles in tumorigenesis and tumor progression. This family has several gene members. c-Myc has been well studied compared to other members and was first discovered as a homolog of v-Myc oncogenes of the avian myelocytomatosis virus [52]. c-Myc contains a basic (i.e., cationic) region that determines the sequence-specific DNA binding, followed by the helix–loop–helix leucine zipper, known as DNA protein binding [52]. c-Myc interacts with Max to fully become an active transcription factor [53]. The active heterodimer binds to the consensus sequence CACGTG on c-Myc’s target genes to activate their transcriptions [54,55]. Unlike c-Myc, Max can homodimerize itself, but it prefers forming a heterodimer with c-Myc. The Max homodimer is not an active transcription factor. In normal cells, c-Myc undergoes ubiquitination and subsequently is degraded by the proteasome, so it is expressed at a low level in the cytoplasm [56]. c-Myc is overexpressed in 40% of tumors due to increased transcription and protein stability [56]. Mutant K-Ras G12V, one of the most common mutated oncogenes driving cancers, increases c-Myc protein levels in pancreatic cancer cells, primarily through translational and post-translational processes [57]. c-Myc has been demonstrated to function as a transcriptional factor, activating several genes that take part in the Warburg effect, glutamine consumption, and lactate production in cancer cells. The c-Myc target genes that are relevant to “aerobic glycolysis” in cancer cells include almost all the enzymes in the glycolytic pathway as well as LDH-A [58].

### 3.6. Transporters Integral to “Aerobic Glycolysis” in Cancer Cells

#### 3.6.1. Glucose Transporters

To support the unrestricted growth of cancer cells, several nutrient transporters are upregulated to promote the influx of essential nutrients to feed into various metabolic pathways that are reprogrammed in the cells [59,60,61]. Among them, glucose transporters are integral to the Warburg hypothesis and the associated “aerobic glycolysis”. The flux of glucose through glycolysis is accelerated in cancer cells. This necessitates an increased influx of glucose from circulation into these cells. The first glucose transporter that was found to be upregulated in the plasma membrane of cancer cells was GLUT1, also identified as SLC2A1 [62,63]. In some cancers, GLUT3 (SLC2A3) is also upregulated. SLC2A1 and SLC2A3 belong to the class of facilitative glucose transporters with no involvement of ion gradients in the transport process. The genes coding for these two transporters are transcriptional targets for HIF-1α [62,63,64]. The Michaelis constant for this transporter for glucose is ~3 mM, which is close the normal plasma level of glucose (~5 mM). SLC2A1 is not sensitive to insulin. Even though there is no energy involved in the transport mechanism of SLC2A1 and SLC2A3, the increased density of these transporters in the plasma membrane due to increased HIF-1α-induced transcription results in an increased influx of glucose from the circulation into cancer cells to feed into the glycolytic pathway.

The increased flux of glucose into cancer cells compared to normal cells forms the basis of positron emission tomography (PET) as a diagnostic tool for detecting tumors in the body in vivo. PET scan is a method of noninvasive imaging that merges the biochemical energy utilization variance of different cell types to the diagnosis of pathology [65]. While it can be used for any tissue that has a regional variation in glucose uptake, such as the brain, muscle, kidney, etc., this imaging technique has become an important diagnostic tool in the oncology field not only to detect tumors but also to evaluate the efficacy of chemotherapy or immunotherapy [66]. The biochemical basis of the success of this technique is directly related to the Warburg effect. The process of PET scan imaging uses the acceptance of 2-deoxy-D-glucose as a substrate for SLC2A1. Since cancer cells express higher levels of this transporter than the normal cells in areas surrounding the tumor, 2-deoxy-D-glucose, when injected into blood, gets into cancer cells at several-fold higher levels than into surrounding normal cells. Once inside the cells, this glucose analog gets phosphorylated by hexokinase in the first step of the glycolytic pathway to generate 2-deoxy-D-glucose-6-phosphate. However, unlike glucose-6-phosphate, which goes through subsequent steps in glycolysis, 2-deoxy-D-glucose-6-phosphate cannot enter the second step in the pathway, thus leading to accumulation in cells. This preferential accumulation in cancer cells versus normal cells can be detected by the PET scan if 2-deoxy-D-glusose is labeled with a positron emitter, hence the use of ^18^F-2-deoxy-D-glucose (^18^FDG) as the PET probe in which ^18^F is the positron emitter. This enables the PET scan to tag the exact location of the tumor in vivo.

For a long time, it was thought that only SLC2A1 is responsible for the increased influx of glucose into cancer cells and that SLC2A3 may contribute to glucose uptake in cancer cells to some extent. However, it was discovered later that certain tumors also upregulate the Na^+^-coupled glucose transporters SGLT1 (SLC5A1) and SGLT2 (SLC5A2) [67]. At the functional level, the increased expression of SLC5A2 has been demonstrated unequivocally [67,68]. In contrast to SLC2A1 and SLC2A3, SLC5A2 is a concentrative transporter and thus has a greater efficiency than SLC2A1/SLC2A3 in transporting glucose into cancer cells. Interestingly, 2-deoxy-D-glucose is not recognized as a substrate by SLC5A2. Therefore, the reliability of a PET scan with ^18^FDG as the probe in cancer detection still depends solely on SLC2A1 (and SLC2A3). α-Methyl-D-glucopyranoside is a selective substrate for SLC5A2; it is not recognized by SLC2A1/SLC2A3. ^18^F-labeled α-methyl-4-deoxy-D-glucopyranoside has been shown to be effective as the PET probe for detecting tumors that are positive for SLC5A2 [67,68]. This may have an advantage over ^18^FDG for SLC5A2-positive tumors to differentiate them from the surrounding normal tissues because SLC2A1 is ubiquitously expressed whereas SLC5A2 expression is restricted almost exclusively to the kidney among normal tissues.

#### 3.6.2. Lactate/H^+^ Symporters (Monocarboxylate/H^+^ Cotransporters)

Lactic acid is the end product of “aerobic glycolysis” in cancer cells. If this acid is not removed from the cells, it will lead to intracellular acidification with consequent detrimental effects on cell growth and proliferation. The ideal candidate transporters for the removal of lactic acid from cells are monocarboxylate transporters (MCTs) that mediate the symport of lactate and H^+^ in an electroneutral mechanism. These transporters belong to the solute carrier 16 gene family (*SLC16*). There are 14 members in this family; among them, 4 members, MCT1 (SLC16A1), MCT2 (SLC16A7), MCT3 (SLC16A8), and MCT4 (SLC16A3), function as lactate/H^+^ symporters [69]. The transport process mediated by these four MCTs is bidirectional, the direction of lactate/H^+^ movement being dictated simply by the direction of the gradient for lactate. Among these four MCTs, only MCT1 (SLC16A1) and MCT4 (SLC16A3) have been shown to be most relevant to cancer cells [3,4,70,71].

MCT1 (SLC16A1) and MCT4 (SLC16A3) are highly upregulated in cancers. c-Myc, Wnt signaling, NF-κB, and mutant p53 are transcriptional inducers of MCT1 [4,71,72]. Hypoxia induces the expression of MCT4 through an HIF-1α-dependent mechanism [4,71,73]. MCT1 exhibits a higher affinity for lactate compared to MCT4 and is kinetically more suited to mediate the influx of extracellular lactate into cells. In contrast, MCT4, which has a relatively lower affinity for lactate, is more suited to mediating the efflux of lactate from cancer cells that generate massive amounts of this metabolite intracellularly. Accordingly, these two transporters are not upregulated uniformly in all cancer cells within the tumor. Tumor cells consist of different subpopulations, such as oxidative and glycolytic tumor cells. Oxidative tumor cells are located close to blood vessels and receive enough oxygen, whereas glycolytic tumor cells are far from blood vessels and hence are exposed to hypoxic conditions. This represents a significant revision to the original Warburg hypothesis in which all cancer cells were assumed to uniformly undergo “aerobic glycolysis”. This recent discovery of heterogeneous populations of cancer cells in terms of oxygen exposure has introduced a new term, the “reverse Warburg effect” [74,75]. This term describes the use of extracellular lactate for energy production via mitochondrial oxidation in some populations of cancer cells that are exposed to normal oxygen. This does not necessarily mean that the Warburg effect occurs only in hypoxic cancer cells; if that were the case, what happens in such cells will be “anaerobic glycolysis”, and not “aerobic glycolysis” as defined by the Warburg effect. A significant fraction of oxygen-exposed cancer cells still undergo the Warburg effect with “aerobic glycolysis”. It is likely that, among the oxygen-exposed cancer cells, which are subject to the Warburg effect and which are subject to the reverse Warburg effect is dependent on the expression levels of the oncogenes c-Myc, HIF-1α, and mutant p53, relative activities of LDH-A versus LDH-B, and the levels of other regulatory mechanisms that facilitate “aerobic glycolysis”. As a consequence, lactate is shuttled between the two different subpopulations of cancer cells within the tumor (Figure 2). In glycolytic tumor cells undergoing the Warburg effect, lactate is produced from pyruvate by LDH-A and exported into the tumor microenvironment by MCT4. In oxidative tumor cells undergoing the reverse Warburg effect, extracellular lactate is taken up by MCT1 and gets converted to pyruvate by LDH-B for subsequent energy generation in mitochondria. This lactate shuttling may not be limited to cancer cells within the tumor. Activated T cells perform “aerobic glycolysis” and produce lactate and release it into the external environment. However, this MCT4-mediated lactate-release mechanism is impaired in tumors because of the presence of a high concentration of lactate in the tumor microenvironment (Figure 2). Consequently, the proliferation of cytotoxic T cells is suppressed, hence leading to the tumor’s ability to evade attack by the immune system. In contrast, tumor-associated endothelial cells and macrophages take up lactate from the tumor microenvironment via MCT1 for energy production. This promotes cell proliferation and tumor angiogenesis in endothelial cells and also induces macrophage polarization to generate pro-tumor M2 macrophages involved in immunosuppression and neovascularization (Figure 2). When these endothelial cells proliferate during vasculogenesis, they differentiate into “stalk” cells and “tip” cells, which opt for aerobic glycolysis and attain features such as increased motility and invasion associated with the sprouting of nascent blood vessels. 

#### 3.6.3. Additional Transporters for H^+^ Export in Cancer Cells

Cancer cells utilize several other mechanisms to prevent intracellular acidification caused by lactic acid production. Many of these transporters export H^+^ out of the cells and their expression is induced by multiple oncogenic drivers [76]. Such transporters include Na^+^/H^+^ exchanger-1 (NHE-1/SLC9A1) [77], Na^+^/HCO_3_ cotransporter NBCn1 (SLC4A7) [78], and V-type H^+^ pump [79]. Cancer cells also use carbonic anhydrase IX in the regulation of intracellular pH. Recent studies have shown that the amino acid transporter SLC38A5 is upregulated in some cancers [80,81]. This is a unique amino acid transporter that functions as an amino-acid-dependent Na^+^/H^+^ exchanger with intracellular alkalinization in the presence of extracellular amino acid substrates [82]. As such, the upregulation of SLC38A5 in cancer cells has a dual role, namely the provision of amino acids to support cell growth and proliferation and export of H^+^ to prevent intracellular acidification.

### 3.7. Acidic pH in Tumor Microenvironment and Its Relevance to Tumor Growth

#### 3.7.1. Non-Specific Effects of Acid pH on Tumor Microenvironment

With the concentration of lactic acid in the range of 30–40 mM in the extracellular space, cancer cells and the stromal cells in the tumor microenvironment are bathed in a medium with an unphysiological acid pH (pH 6.0–6.5). Cancer cells survive this acid pH because they upregulate their biochemical machinery to protect themselves from the harmful effects of extracellular acid pH with the aid of oncogenes and consequent changes in the expression levels of various transporters and biochemical processes. In contrast, the stromal cells in the same acidic environment do not have this luxury and hence face the consequences. In addition, cancer cells and cancer-cell-associated stromal cells also begin to secrete various proteases, primarily metalloproteinases, which start digesting the extracellular proteins such as collagen. As such, the negative effects of acidic pH on stromal cells and the clearance of extracellular connective tissue pave the way for cancer cells to grow further.

In addition, the acidic pH in the extracellular medium coupled with the efficient maintenance of intracellular pH in cancer cells creates an inwardly directed H^+^ gradient across the plasma membrane in these cells. There are several transporters for important nutrients whose transport process is fueled by a transmembrane H^+^ gradient [83,84,85], and most them are upregulated in cancer cells to exploit the now naturally occurring pH gradient across their plasma membrane to energize the transfer of these nutrients into cancer cells to support their growth and proliferation. The first example is MCT1, which has already been discussed. This transport activity of MCT1 to mediate the influx of extracellular lactate into oxidative cancer cells is activated by the inwardly directed H^+^ gradient. While lactate is the “waste” product of cellular metabolism in cancer cells with “aerobic glycolysis”, it is an energy-rich metabolite for oxidative cancer cells. The efficient uptake of lactate via MCT1 with subsequent mitochondrial oxidation generates ATP to fuel cancer growth. The substrates of other H^+^-coupled nutrient transporters are also very important for cell proliferation. This includes peptides, amino acids, folate, iron, and citrate.

#### 3.7.2. H^+^-Coupled Peptide Transporter PEPT1 (SLC15A1)

PEPT1 transports small peptides consisting of two to three amino acids and it represents the first H^+^-coupled nutrient transporter discovered in the mammalian cell plasma membrane [86,87]. This transporter is upregulated in cancers [88,89,90,91]. Normal plasma contains small peptides only at low levels, but this is most likely not the case in the tumor microenvironment. The metalloproteinases secreted by tumor cells and stromal cells digest extracellular proteins and release small peptides in the local environment, and these peptides are the likely substrates for PEPT1, which is upregulated in cancer cells [91]. The naturally occurring pH gradient across the plasma membrane of these cells would energize the transport activity of PEPT1 to satisfy their amino acid nutrition.

#### 3.7.3. H^+^-Coupled Amino Acid Transporter PAT1 (SLC36A1)

PAT1 transports small amino acids such as proline and glycine [92]. Even though the expression of this transporter is not altered in cancer, its expression is evident [88]. This could be relevant to amino acid nutrition in cancer cells because proline and glycine are likely to be present in the tumor microenvironment at higher concentrations than in normal plasma because of the metalloproteinase-mediated degradation of extracellular collagen, a protein rich in these two amino acids. Recent studies have shown that the transport activity of PAT1 is coupled to the activation of mTORC1 signaling [93], thus the transporter becoming relevant to tumor growth.

#### 3.7.4. H^+^-Coupled Folate Transporter PCFT (SLC46A1)

Folic acid is an obligatory vitamin for the synthesis of purines, pyrimidines, and nucleotides, and mammalian cells employ multiple transport mechanisms as well as receptor-mediated endocytosis to acquire this vitamin from circulation. One of the transport mechanisms involves the H^+^-coupled transporter PCFT (SLC46A1) [94,95]. This transporter is highly expressed in cancers and is being exploited to deliver cytotoxic folate analogs to antagonize the biological functions of folate in cancer chemotherapy [96].

#### 3.7.5. H^+^-Coupled Divalent Metal Ion Transporter DMT1 (SLC11A2)

Iron is a micronutrient obligatory for many vital functions necessary for the survival and proliferation of cells. It is an integral component, both in the free form and also in the form of heme, in the electron transport chain for ATP production. It is also an obligate cofactor for numerous enzymes and for hemoglobin. Most cells acquire this micronutrient in the form of transferrin-bound iron via receptor-mediated endocytosis involving the transferrin receptor. But, free unbound iron is also transported into cells across the plasma membrane via the H^+^-coupled divalent metal ion transporter DMT1 (SLC11A2) [97]. Interestingly, the same transporter is required to deliver free iron from lysosomes into cytoplasm following the entry and lysosomal processing of transferrin-bound iron in receptor-mediated endocytosis [98]. In both cases, the transport process is dependent on a transmembrane H^+^ gradient. SLC11A2 is upregulated in cancers [99], thus providing an efficient mechanism for cancer cells to accumulate iron in support of their growth and proliferation.

#### 3.7.6. Na^+^/H^+^-Coupled Citrate Transporter NaCT (SLC13A5)

Citrate is a key metabolic intermediate at the junction of important metabolic pathways. It is at the center of the citric acid cycle, functions as a regulator of glycolysis, and serves as the carbon source for fatty acid synthesis. For a long time, it was thought that mitochondrial generation is the sole source of citrate in cells. But, with the discovery of a plasma membrane transporter selective for citrate [100,101], it has become clear that citrate in circulation could form a significant source of citrate for cells. Extracellular citrate fuels cancer growth [102,103]. This transporter is upregulated in some cancers, particularly liver cancer [104]. Even though it is a Na^+^-coupled transporter [100,101], its activity is markedly stimulated by extracellular acidic pH [105]. The transporter possesses more than one Na^+^-binding site and it seems that one or more of these binding sites have a much higher affinity for H^+^ than for Na^+^. As such, SLC13A5 actually functions as a Na^+^/H^+^-coupled citrate transporter. Citrate is present at significant levels in normal circulation (~200 μM); therefore, SLC13A5 has the ability to deliver citrate into cancer cells and its transport activity is stimulated by the acidic pH in the tumor microenvironment. This phenomenon might be even more relevant to tumors that grow in bone after metastasis because bone holds more than 60% of citrate in the body in its matrix. When tumors grow in bone in the form of an “osteoclastic”-type metastasis, the bone matrix is degraded and releases citrate in free form, which could form an important source of this metabolite to fuel the growth of tumors.

### 3.8. Lactate as a Signaling Molecule: Lactate Receptors

Lactate has always been considered as an interesting metabolite because of certain unique features. It is the sole end product of glycolysis in mature erythrocytes, which use glucose as the principal energy source. Due to a lack of mitochondria, pyruvate generated at the end of the glycolytic pathway has to be converted into lactate to recycle NAD^+^. It is also at the center of the Cori cycle, a metabolic crosstalk between skeletal muscle and liver. During exercise, skeletal muscle needs more oxygen than what is readily available, thus being exposed to hypoxia. This results in anerobic glycolysis during exercise, thus generating lactate as the end product of glycolysis, which is then released into circulation. This explains why plasma levels of lactate rise during exercise. Liver picks up this lactate to use it as a carbon source for gluconeogenesis, and the newly synthesized glucose is then released into circulation. This glucose–lactate–glucose cycle is called the Cori cycle. Based on these biochemical features, lactate is always considered as the biomarker of oxygen deficit. This is further supported by the fact that ischemic tissues generate lactate as the end product of glucose metabolism. But, recently, this ubiquitous metabolite has come to the forefront of cancer biology as a signaling molecule. This elevation of lactate from the old status of “waste product of glycolysis” to the new status of “hormone” has renewed interest in this molecule in the field of cancer biology. When the biological outcomes of lactate signaling are considered, a rationale emerges as to why this metabolite is a biomarker of oxygen deficit. These outcomes include, among many other things, the stabilization of HIF-1α protein and promotion of angiogenesis. These lactate-induced processes are directly related to the promotion of tumor growth and metastasis, thus underscoring the biological importance of lactate as the principal metabolite of glucose metabolism in cancer cells. 

#### 3.8.1. Intracellular Lactate Receptor NDRG3

NDRG3 is one of the four members of the NDRG (N-Myc-downstream regulated genes) family of proteins. NDRG3 functions as a tumor promoter [106,107]. It binds to c-Raf and activates ERK1/2 signaling, which regulates gene transcription, cell proliferation, migration, differentiation, and cytoskeletal remodeling to support tumor growth in the late stage of hypoxia [108,109]. NDRG3 is subject to proteasomal degradation and this requires its binding to VHL. Interestingly, lactate interferes with the interaction of NDRG3 with VHL by directly binding to NDRG3, thus preventing the depletion of NDRG3 via proteasomal degradation [108]. Thus, NDRG3 functions as an intracellular receptor for lactate. This is in addition to the influence of lactate on prolyl hydroxylate-2 (PHD2), which hydroxylates NDRG3 for subsequent binding to VHL and consequent proteasomal degradation. It is also possible that the binding of lactate to NDRG3 is needed not only for the stabilization of the NDRG3 protein but also for enabling the binding of NDRG3 to c-Raf. This lactate-stimulated NDRG3-Raf-ERK signaling promotes angiogenesis and cell growth.

#### 3.8.2. Cell-Surface G-Protein-Coupled Receptor GPR81 for Lactate

Lactate also elicits its intracellular signaling by functioning in the extracellular milieu as an agonist for the G-protein-coupled receptor GPR81 on the plasma membrane [110,111]. This receptor was first identified in adipocytes where its activation by lactate reduces intracellular levels of cAMP, which reduces the hydrolysis of triglycerides by inactivating the hormone-sensitive lipase. Of importance to the field of cancer are the findings that GPR81 is expressed at high levels in cancer and that the receptor functions as a tumor promoter [6,112]. This has direct relevance to tumor growth because cancer cells generate lactate and release it into the tumor microenvironment where it can function extracellularly as an agonist for GPR81. As many other cell types in the stroma surrounding the tumor also express this receptor, lactate present in the extracellular milieu functions as an GPR81 agonist not only in cancer cells but also in adjacent non-cancer cells.

#### 3.8.3. Autocrine Functions of GPR81/Lactate in Tumor Growth

Autocrine signaling involves a hormone secreted by a given cell binding to a cell-surface receptor on the same cell to induce downstream effects. Cancer cells secrete lactate and also express the lactate receptor GPR81, thus paving the path for autocrine signaling (Figure 3). Downregulation of GPR81 in breast cancer cells decreases tumor invasion and migration [113,114]. This effect is associated with a decreased expression of three critical enzymes related to aerobic glycolysis: hexokinase 2, PFK-1, and LDH-A. As such, lactate elicits a positive feedback regulation on its own production by acting on GPR81 in an autocrine manner. In addition, the intracellular signaling resulting from GPR81 activation by lactate leads to the promotion of synthesis and secretion of pro-angiogenic factors amphiregulin, platelet-derived growth factor, serpin peptidase inhibitors Serpin E1 and Serpin F1, plasminogen activator, and vascular endothelial growth factor. This most likely occurs via activation of the PI3K/ATK pathway. In pancreatic cancer cells, knockdown of GPR81 reduces cell survival, mitochondrial activity, and monocarboxylate transporters MCT1 and MCT4 [115]. Lactate stabilizes HIF-1α via a GPR81-mediated decrease in cAMP and consequent inhibition of protein kinase-A in B16-F10 and Hepa1-6 cells [116]; this is in addition to the ability of lactate to inhibit the prolylhydroxylation of HIF-1α and prevent its proteasomal degradation. The activation of GPR81 by lactate also helps tumor cells to escape from cytotoxic immune cells. In human lung cancer cells, activated GPR81 reduces intracellular cAMP levels and inhibits protein kinase-A, which results in the activation of the transcriptional coactivator TAZ [117]. TAZ then interacts with TEAD1 and the resultant complex causes the transcriptional activation of the gene coding for PD-L1 (programmed cell death ligand 1). PD1 is an inhibitory receptor expressed in T cells during activation. The binding of PD-1 (T cell) and PD-L1 (cancer cell) inhibits T cell activation and prevent T cells from killing cancer cells. Thus, inducing PD-L1 on cancer cells’ surface facilitates cancer cells to escape the immune checkpoints. Moreover, GPR81 signaling induces DNA repair proteins such as BRCA1, nibrin, and DNA-dependent protein kinases in cervical cancer cells [118]. It also enhances doxorubicin chemoresistance by increasing the expression of ABCB1, a drug efflux transporter. Interestingly, a recent study has reported that lactate released by cancer cells initiates a signaling pathway via GPR81 to induce the expression of the same receptor to a higher level to promote the autocrine signaling even further [119]. This process involves the induction of the expression of the transcription factor Snail and promotion of the formation of the Snail/EZH2/STAT3 complex, which then directly binds to the promoter of GPR81 gene to induce expression. In another study, lactate generated by breast cancer cells enhances their growth and invasiveness in an autocrine manner through regulation of extracellular matrix properties and Notch signaling [120]. 

#### 3.8.4. Paracrine Functions of GPR81/Lactate in Tumor Growth

Paracrine signaling involves the secretion of a hormone by a given cell with subsequent action of the hormone on a cell-surface receptor expressed on a different cell present in an adjacent or distant location to elicit a signal. A recent report by Brown et al. [121] constitutes a prime example for the paracrine function of lactate in breast cancer. Lactate generated and released by breast cancer cells activates GPR81 present in dendritic cells to suppress antigen presentation via downregulation of MHC-II. This aids tumor cells in evading immunosurveillance because the dendritic cells are now defective in handling and presenting tumor-cell-specific antigens to cytotoxic T cells. The cancer-cell-generated lactate also acts on intra-tumoral plasmacytoid dendritic cells to suppress the production of cytokines that are needed for the proliferation of T cells [122]. At the same time, the secretion of the tryptophan metabolite kynurenine is also increased in these dendritic cells in response to GPR81 activation by lactate, which then induces the production of tumor-suppressive regulatory T cells. The combination of interference with T cell expansion and promotion of Treg production enhances the ability of tumor cells to escape immune cells. Collectively, these studies show that lactate generated by cancer cells promotes immune evasion of the tumor [123]. Similarly, lactate released from inflammatory bone marrow neutrophils binds to GPR81 on endothelial cells and subsequently induces vascular endothelial cadherin that increases blood vessel permeability and neutrophil mobilization [124]. This may have relevance to inflammation-associated cancers such as colon cancer and even breast cancer if cancer-cell-generated lactate elicits a similar effect on tumor-associated endothelial cells. Thus, the paracrine function of GPR81/lactate plays various roles in cell–cell communication relevant to cancer growth (Figure 3).

### 3.9. Downstream Metabolic Consequences of Aerobic Glycolysis in Cancer Cells

#### 3.9.1. Pentose Phosphate Pathway and Antioxidant Machinery

A robust antioxidant machinery is important for the survival and also chemoresistance of cancer cells. Catabolic metabolism is oxidative and the mitochondrial electron transport chain generates reactive oxygen species all the time because a small portion of molecular oxygen that is used in this process does not go through complete reduction to water but gets out of the electron transport chain as partially reduced. For complete oxidation of O_2_ to form water, four electrons need to be added. The addition of fewer than four electrons to O_2_ results in reactive oxygen species: superoxide resulting from one electron and peroxide resulting from two electrons. Unless these reactive oxygen species are removed, cells face the risk of oxidative damage and cell death. The removal of superoxide and peroxides involves the glutathione/glutathione peroxidase/glutathione reductase system, which needs NADPH as the cofactor. There are two major pathways that generate NADPH: the pentose phosphate pathway and malic enzyme. Both pathways are activated in cancer [26]. When aerobic glycolysis occurs at a much faster rate in cancer cells, the cellular levels of various intermediates in the pathway also go up. Glucose-6-phosphate, one of the intermediates, is the starting point for the pentose phosphate pathway. The rate-limiting enzyme in this pathway is glucose-6-phosphate dehydrogenase, which is induced by c-Myc [125]. The increased availability of the starting material and the increased activity of the rate-limiting enzyme fuel the increased flux of glucose-6-phosphate through the pentose phosphate pathway to generate NADPH. Another product of this pathway is ribose-5-phosphate, which promotes purine and pyrimidine synthesis because of the production of phosphoribose pyrophosphate, a substrate as well as an activator of nucleotide synthesis, which is necessary for rapidly proliferating cells such as cancer cells. Thus, the increased activity of aerobic glycolysis is tied to an increased activity of the pentose phosphate pathway. Malic enzyme converts malate into pyruvate along with the production of NADPH. This enzyme is also upregulated in cancer cells [126]. The resultant robust antioxidant machinery protects cancer cells from oxidative damage.

This machinery is also related to the development of chemoresistance. Many chemotherapeutic agents (e.g., cisplatin) are inactivated by reactions that use glutathione. When cellular levels of NADPH are high, the levels of reduced glutathione are also kept high because of the efficient conversion of oxidized glutathione into reduced glutathione. Therefore, the robust antioxidant machinery detoxifies chemotherapeutic drugs, thus favoring drug resistance and protecting the cancer cells from cell death induced by such drugs.

Glutathione is an obligatory component of the antioxidant machinery in cancer cells. It is synthesized using three amino acids: glutamate, cysteine, and glycine. Among these three amino acids, cysteine is rate-limiting. This amino acid is provided to cancer cells in the form of cystine via the transporter xCT/SLC7A11, which is upregulated in cancer cells [27,28]. This upregulation is related at least partly to the ability of the tumor suppressor protein p53 to suppress the expression of SLC7A11 and also to the discovery that p53 is a heme-binding protein [127]. Interestingly, this phenomenon is also related to ferroptosis, an iron-dependent non-apoptotic cell death process. Cancer cells accumulate iron to support their rapid metabolic machinery, but, at the same time, these cells are obligated to protect themselves from ferroptosis. This is accomplished by the increased levels of heme in cancer cells resulting from increased iron, followed by the binding of heme to p53 and the resultant p53-heme complex undergoing proteasomal degradation. The net result is the depletion of p53 in iron-loaded cancer cells with a consequent increase in SLC7A11 expression. Ferroptosis involves iron-induced lipid peroxidation, and glutathione is a potent blocker of this process. As such, the increase in the expression and activity of SLC7A11 in cancer cells brings in cystine from circulation to promote glutathione synthesis, thus providing a mechanism for cancer cells to protect themselves from iron-induced cell death. Chronic exposure of pancreatic cancer cells to excess iron induces epithelial-to-mesenchymal transition, causes a loss of p53, and consequently increases SLC7A11 expression [128]. Similarly, the pharmacological inhibition of SLC7A11 in triple-negative breast cancer cells reduces cell proliferation and also suppresses growth of these cells in mouse xenografts [129]. 

#### 3.9.2. Serine Biosynthesis and One-Carbon Metabolism

Serine is an important source of one-carbon moieties for one-carbon metabolism. This pathway begins with the generation of *N*^5^, *N*^10^-methylenetetrahydrofolate from the conversion of serine into glycine by the reaction mediated by serinehydroxymethyl transferase. *N*^5^, *N*^10^-methylene tetrahydrofolate can then be converted into *N*^5^-formyltetrahydrofolate and *N*^5^-methyltetrahydrofolate. These folate derivates serve as the source of one-carbon moieties in one-carbon transfer reactions that participate in multiple metabolic processes, including purine and pyrimidine synthesis, nucleotide synthesis, and homocysteine metabolism. Therefore, serine becomes an important amino acid for cancer cells. Even though many amino acid transporters that can transfer extracellular serine into cells are upregulated in cancer cells [12,13,14,15], the endogenous synthesis of serine is also activated to satisfy the increased demand for this amino acid. The starting material for serine synthesis is 3-phosphoglycerate, an intermediate in glycolysis. The first and rate-limiting enzyme in serine biosynthesis is phosphoglycerate dehydrogenase, which is induced in cancer cells [130].

#### 3.9.3. Glutaminolysis and Glutamine Transporters

Similar to serine, the need in cancer cells for glutamine is also very high. Glutamine serves as the nitrogen source for purine synthesis, provides glutamate for glutathione synthesis, and also activates mTORC1. In addition, it feeds into the citric acid cycle to generate ATP and also the signaling metabolite lactate. The conversion of glutamine to lactate involves a series of reactions, collectively called “glutaminolysis” (Figure 4). This pathway utilizes a truncated form of the citric acid cycle and generates NADH and FADH_2_ that can be used in ETC/OXPHOS to produce ATP, malate that can be used by malic enzyme to generate NADPH to boost the antioxidant machinery, and pyruvate that can be converted to lactate, the tumor-promoting signaling molecule. It has been estimated that almost one-half of total lactate produced in cancer cells originates from glutaminolysis, the remainder being generated in “aerobic glycolysis” [131]. A portion of this malate also undergoes the next step in the citric acid cycle to generate NADH, which is then used in ETC/OXPHOS to produce additional ATP. This underscores the critical role of glutamine metabolism in cancer, particularly when given the notion that existed for a long time that almost all lactate generated in cancer cells comes from aerobic glycolysis.

There are several amino acid transporters that can transport glutamine. This list includes ASCT2 (SLC1A5) [132], LAT1 (SLC7A5) [133], ATB^0,+^ (SLC6A14) [134,135,136,137], and SN2 (SLC38A5) [80,81]. SLC1A5, when originally cloned, was thought to be the well-characterized amino acid transport system B^0^ [138], but was later found to be one of the isoforms of the amino acid transport system ASC (alanine–serine–cysteine transporter), now identified as ASCT2 [139]. This transporter functions as an obligatory amino acid antiporter, meaning that when one of its substrates enters the cell, another substrate of the transporter leaves the cell. Interestingly, SLC1A5 transports glutamine and its expression in cancer cells is induced by c-Myc [140]. There is also evidence for a functional crosstalk between SLC1A5 and SLC7A5; both these transporters being obligatory amino acid exchangers, glutamine entering the cell via SLC1A5 is exchanged for extracellular leucine via SLC7A5 [141]. Since leucine is an activator of mTORC1, this functional interaction between the two transporters potentiates mTOR signaling and hence cell growth and proliferation. SLC6A14 is induced in cancer cells via Wnt signaling [142]. SLC38A5 is induced in cancer cells by c-Myc [140]. As such, multiple amino acid transporters satisfy the addiction of cancer cells to glutamine and the process is coupled to oncogenic signaling. 

The first step in glutaminolysis is the conversion of glutamine to glutamate, catalyzed by glutaminase. There are two isoforms of this enzyme, kidney-specific and liver-specific. The former, also called GLS1, functions as a tumor promoter as its expression is upregulated in cancer, and this process is mediated by c-Myc [143]. 

#### 3.9.4. Reductive Carboxylation and Fatty Acid Synthesis

One of the reactions in the citric acid cycle within mitochondria is the conversion of isocitrate to α-ketoglutarate, which is mediated by isocitrate dehydrogenase-3 (IDH-3) and involves the generation of CO_2_ and NADH. Since this reaction catalyzes decarboxylation (removal of CO_2_ from isocitrate) and oxidation (removal of electrons from isocitrate to convert NAD^+^ to NADH), it is called “oxidative decarboxylation”. Mitochondria also contain IDH-2, another isoform of the same enzyme. This enzyme can use an NAD^+^/NADH pair or NADP^+^/NADPH pair as cofactors. When cellular levels of NADPH are high as in cancer cells, IDH-2 can perform the reverse reaction in which α-ketoglutarate gets converted to isocitrate. This reaction involves the carboxylation as well as addition of electrons (NADPH gets converted to NADP^+^). Accordingly, this reaction is called “reductive carboxylation” (Figure 4). Cancer cells perform this reaction to reverse the citric acid cycle to generate citrate from α-ketoglutarate. Glutamine is the source of this α-ketoglutarate (glutamine → glutamate → α-ketoglutarate). Citrate is then converted to acetyl-CoA by ATP-citrate lyase and the resulting acetyl-CoA is then used as the carbon source for fatty acid synthesis. Increased fatty acid synthesis is necessary for cancer cell proliferation, especially for membrane biogenesis.

#### 3.9.5. Impact of Amino Acid Transporters in Cancer Cells on Tumor-Associated Immune Cells: Concept of Immunological Synapse

Immunological synapse is a specialized cellular junction between cancer cells and cytotoxic lymphocytes (T lymphocytes as well as natural killer cells). In the case of T lymphocytes, the synapse is formed when the T cell receptor engages with the major histocompatibility complex on the surface of cancer cells. Once activated by this interaction, the T cells undergo rapid proliferation, and the resultant cytotoxic T cells then kill cancer cells. This represents an important facet of immune surveillance and anti-tumor immunity. Like any rapidly proliferating cell, activated T lymphocytes have an obligatory need for essential amino acids for their rapid proliferation. When cancer cells upregulate amino acid transporters to satisfy their increased need for amino acids, the levels of amino acids in the extracellular environment, including the immunological synapse, are markedly decreased. This decreases the availability of essential amino acids to activated T lymphocytes and impairs their proliferation. In other words, cancer cells orchestrate the blockade of the expansion of cytotoxic T lymphocytes, thus effectively counteracting the anti-tumor immunity. This phenomenon has been demonstrated for two essential amino acids: tryptophan and methionine. Cancer cells take up tryptophan via SLC7A5 and SLC6A14 and, once inside the cell, tryptophan gets degraded by indoleamine-2,3-dioxygenase-1 (IDO1). This tryptophan-degrading enzyme is upregulated not only in cancer cells but also in antigen-presenting dendritic cells present in tumor-draining lymph nodes. The resultant increase in tryptophan uptake and metabolism in cancer cells and in dendritic cells suppresses the expansion of cytotoxic T lymphocytes. This phenomenon has been the focus of several expert reviews [144,145]. It has to be noted that the blockade of T cell proliferation may not be totally due to tryptophan depletion; it could also involve the metabolites of IDO1-mediated tryptophan breakdown (e.g., kynurenine), which are released from the cancer cells and dendritic cells and elicit detrimental effects on the adjacent T cells with a negative impact on their proliferation. More recently, a similar phenomenon has been shown to occur with regard to methionine, also an essential amino acid [146,147]. Cancer cells upregulate the amino acid transporter SLC43A2, which has a robust ability to mediate methionine uptake. This results in methionine depletion at the immunological synapse, thus affecting metabolism in adjacent T lymphocytes. Methionine is needed not only for protein synthesis but also epigenetic modifications and consequent transcriptional control. When T lymphocytes are subjected to methionine deficiency, their transcriptional landscape is altered, leading to suppression of their proliferation and function. 

Thus, the upregulation of amino acid transporters in cancer cells has a dual role. It provides amino acids to cancer cells to support their growth and proliferation. At the same time, the rapid uptake of amino acids by cancer cells reduces the availability of amino acids to cytotoxic lymphocytes, thus affecting their expansion and function and producing effective immune evasion. 

## 4. Oncometabolites

Oncometabolites refer to small-molecule metabolites that are present at lower levels in normal cells but are increased in cancer cells and also elicit tumor-promoting effects [148,149,150]. These metabolites are generated as a consequence of the rewiring of metabolic pathways in cancer cells. Lactate is considered as an oncometabolite. Cancer cells generate this metabolite at high levels and lactate has several biological functions, GPR81-dependent as well as GPR81-independent, that are involved in the promotion of tumor growth, thus conforming to the definition of an oncometabolite. Citrate may also qualify as an oncometabolite. Two other metabolites commonly included in this category are fumarate and succinate. These are generated in certain cancers at higher-than-normal levels when fumarate hydratase and/or succinate dehydrogenase are defective due to mutations. These metabolites inhibit α-ketoglutarate-dependent dioxygenases, which leads to DNA hypermethylation, thus leading to epigenetic changes conducive for carcinogenesis and cancer growth. D-2-hydroxyglutarate is another oncometabolite that is produced by specific mutants of IDH-1 and IDH-2. These are gain-of-function mutations and the mutated enzymes gain the ability to convert α-ketoglutarate into D-2-hydroxyglutarate. This oncometabolite also functions similar to fumarate and succinate in inhibiting α-ketoglutarate-dependent dioxygenases, thus influencing epigenetics. However, recent studies have discovered an additional function of this oncometabolite [151]. Tumor cells release this metabolite, which is then taken up by T cells, where it inhibits LDH-A, thus reducing the conversion of pyruvate into lactate and consequently forcing oxidative metabolism of pyruvate within mitochondria. Normally, proliferating T cells perform “aerobic glycolysis” similar to cancer cells. When the metabolic phenotype is changed to oxidative metabolism, T cell proliferation is suppressed, which enables tumors to evade immune surveillance. 

## 5. Conclusions

Cancer cells reprogram metabolic pathways to suit their biological needs, and the oncogenic proteins c-Myc, HIF-1α, and mutant p53 aid or initiate this process. Glucose metabolism and glutamine metabolism are the ones that are most affected. The reprogrammed metabolic pathways have led to the coining of new words such as “aerobic glycolysis”, “glutaminolysis” “glutamine addiction”, “reductive carboxylation”, “oncometabolites”, and “reverse Warburg effect” in cancer vocabulary. The entire field of cancer cell metabolism started with the Warburg effect but, over the course of time, has seen considerable expansion and modifications, including the revision of the explanation for the Warburg effect itself. The new information gained in this field has already led to the identification of new therapeutic targets for cancer therapy. This evolution of and shifting in focus to metabolism represent an exciting and refreshing change in recent years in the field of cancer biology. Even though this review focused solely on glucose and amino acid metabolism and the transporters for glucose and amino acids, cancer cell metabolism involves an altered uptake and metabolism of other nutrients as well. This includes fatty acids. Rapid cell proliferation requires an increased availability of fatty acids for energy production as well as membrane biogenesis. Cancer cells satisfy this need by increasing the endogenous synthesis of fatty acids from acetyl-CoA and also by an enhanced uptake of extracellular fatty acids. The endogenous synthesis is potentiated by an increased expression of ATP-citrate lyase that cleaves citrate to generate acetyl-CoA, the building block for fatty acid synthesis [152]. Cancer cells also upregulate fatty acid synthase, which converts acetyl-CoA to long-chain fatty acids [153]. In addition, extracellular fatty acids are taken avidly by cancer cells via multiple transport mechanisms [154]. Taken collectively, it has become clear in recent years that selective nutrient transporters are obligatory to drive the reprogrammed cancer cell metabolism. As such, these transporters represent a novel class of ideal drug targets for cancer treatment. Blockade of the function of these transporters can be achieved with specific small-molecule inhibitors as has been shown with SLC7A11, SLC7A5, and SLC6A14 [155,156,157,158]. Since these transporters are integral membrane proteins in the plasma membrane with the exposure of epitopes on the exoplasmic surface of the membrane, they are also amenable for the design and development of specific monoclonal antibodies that might be able to bind to the transporters and block their transport function.

## Figures and Tables

**Figure 1 cancers-16-00504-f001:**
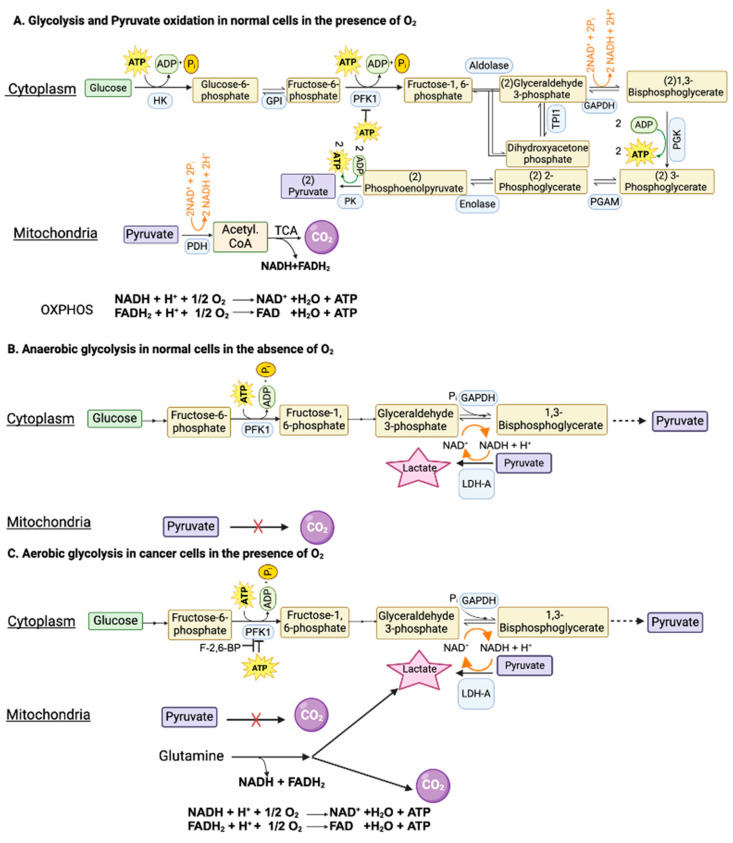
Glycolysis in normal cells and in cancer cells in the presence and absence of oxygen. GAPDH, glyceraldehyde-3-phosphate dehydrogenase; PFK1, phosphofructokinase-1; F-2,6-BP, fructose-2,6-bisphosphate.

**Figure 2 cancers-16-00504-f002:**
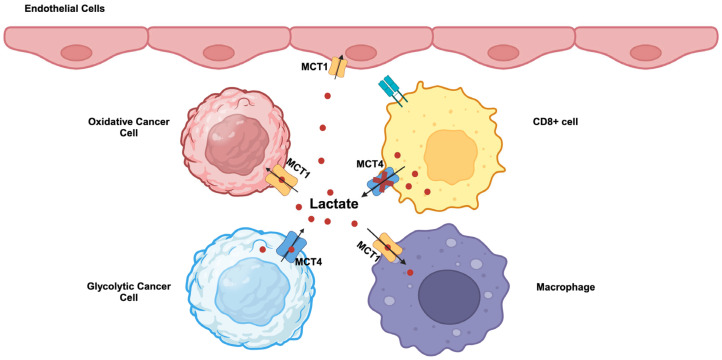
Differential roles of MCT1 and MCT4 in tumors and in non-tumor cells in the tumor microenvironment. MCT1, monocarboxylate transporter 1 (SLC16A1); MCT4, monocarboxylate transporter 4 (SLC16A3); CD8+ cell, cytotoxic T cell.

**Figure 3 cancers-16-00504-f003:**
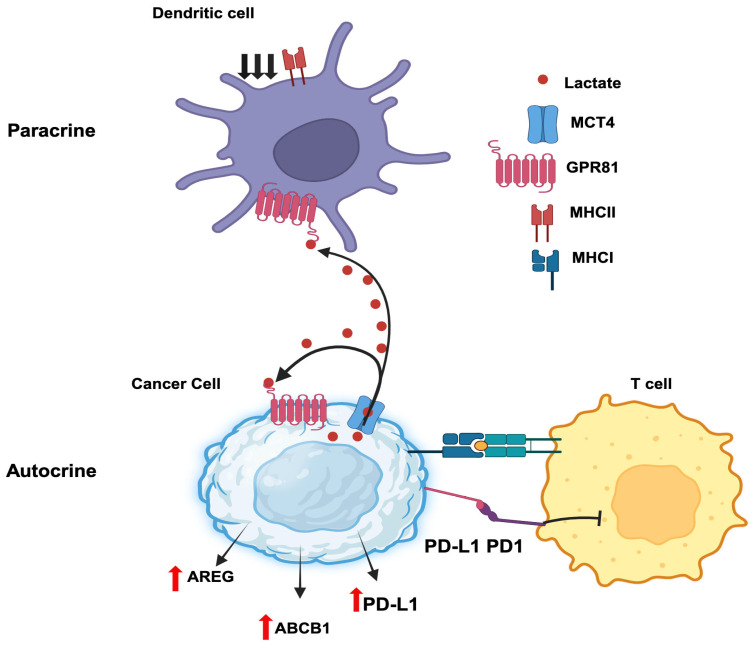
Autocrine and paracrine functions of tumor-cell-derived lactate. AREG, amphiregulin; ABCB1, ATP binding cassette transporter B1; PD1, programmed cell death protein 1; PD-L1, PD ligand 1; GPR81, lactate receptor; MCT4, monocarboxylate transporter 4 (SLC16A3); MHC, major histocompatibility complex.

**Figure 4 cancers-16-00504-f004:**
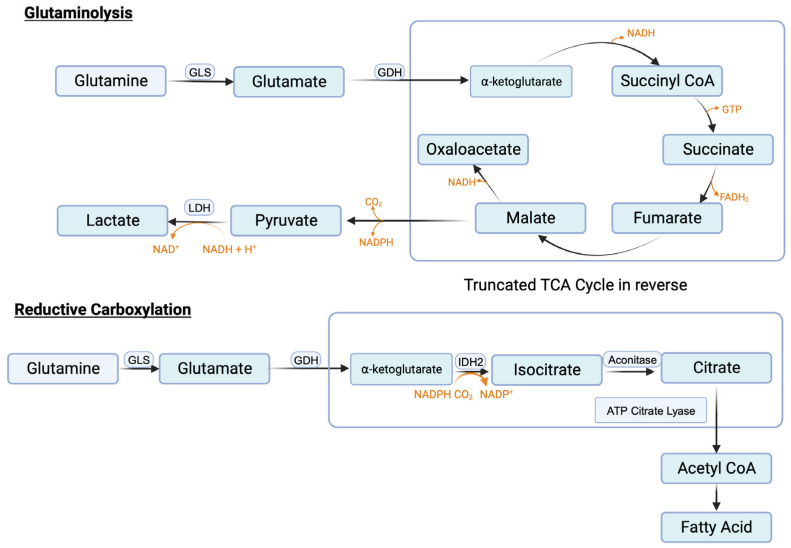
Reactions involved in glutaminolysis and reductive carboxylation.

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
