# Peer review of "Metabolic Signature of Warburg Effect in Cancer: An Effective and Obligatory Interplay between Nutrient Transporters and Catabolic/Anabolic Pathways to Promote Tumor Growth"

_cancers, 2024, doi:10.3390/cancers16030504_

Round 1

Reviewer 1 Report

Comments and Suggestions for Authors

This review article comprehensively summarized the metabolic events related to "Warburg effect".  Overall, this manuscript has a very high quality, and it's very informative and also very important to the field of cell metabolism.  I only have very minor concerns below.

1, It would to helpful to further discuss the role of glutamine transporters and glutaminase in tumor cell growth (especially SLC1A5 and GLS1) since glutamine metabolism is the primary metabolic reprogramming event to compensate "Warburg effect".

2, Another missing point is the pivotal role of GSH biosynthesis pathway as well as cysteine uptake (through transporter SLC7A11) in tumor cell survival.  It would be great to further discuss these parts.   

Comments on the Quality of English Language

This manuscript was well-written and understandable. 

Author Response

This reviewer suggested inclusion of some details on SLC1A5 and glutaminase in terms of their role in cancer and their regulation by oncogenes. We have now added a couple of paragraphs on page 18 of the revised version addressing this comment.

This reviewer also wanted some more details on SLC7A11. We have now done that on page 17 of the revised version.

Reviewer 2 Report

Comments and Suggestions for Authors

I reviewed the manuscript by Mathew et al., entitled “Metabolic Signature of Warburg Effect in Cancer: An Effective and Obligatory Interplay between Nutrient Transporters and  Catabolic/Anabolic pathways to Promote Tumor Growth” this review focusing on the metabolic signature regarding Warburg effect.  The author described in detail the current findings that provided information of the metabolic signature of the Warburg effect. The manuscript is well written and provides interesting information for the reader of the journal. The diagrams and pictures are presented in pest form. Accordingly, the review can be published in the present form without any changes.

Author Response

This reviewer had no comments. He/she was pleased with the original version.

Reviewer 3 Report

Comments and Suggestions for Authors

In this manuscript, the authors summarized the metabolic signature of cancer cells and the role of nutrient transporters in these cells. The focus was on glucose metabolism, lactic acid (the product of aerobic glycolysis in cancer cells), and glutaminolysis within cancer cells. Moreover, the authors discussed how these metabolic processes are regulated and their impact on cancer cells. The conclusion was that understanding the unique metabolic signature of cancer cells can reveal new targets for drugs. The latest information on cancer metabolism and transporters is well-summarized. The following points should be addressed:

1. Regarding transporters, which are central to this paper, it would be beneficial to detail specific results, such as those from knockout experiments. Additionally, organizing the transporters into tables or figures would enhance clarity.

2. The text in all the figures is too small for comfortable reading.

3. In Figure 4, ATP citrate lyase overlaps with the square representing “Truncated TCA cycle in reverse”, which might lead to confusion.

Author Response

This reviewer asked more details on the specific effects of nutrient transporters in cancer cells and also recommended inclusion of the details of these transporters in the form of a Table. We are afraid that such details would be far beyond the scope of the current review. We discuss so many transporters in this article and it would be formidable to include the results of knockout experiments for all these transporters. As evident in the organization of the review, the metabolic pathways constitute the major focus of this article. However, most published reviews ignored the role of transporters in providing the nutrients necessary for these metabolic pathways. This deficiency in most of the current reviews prompted us to discuss the biological significance of such transporters for the cancer cell-specific metabolism. As such, this is not a review solely focusing on transporters in cancer. Nonetheless, we have expanded the review in several places detailing some of the specific aspects of these transporters in response to the comments made by the other three reviewers. We hope that these extra additions to the original version would satisfy this reviewer with regard to this particular comment.

Fig. 1 has been revised by increasing the font size of the letters. The same comment was also made by another reviewer.

The corrections recommended by the reviewer for Fig. 4 have been done. We thank the reviewer for calling our attention to this figure because we found some additional errors that we have now corrected.

Reviewer 4 Report

Comments and Suggestions for Authors

This manuscript provides a comprehensive review of the metabolic reprogramming in cancer cells, extending the discussion beyond the well-established Warburg effect. It emphasizes the interplay between nutrient transporters and metabolic pathways, highlighting the unique metabolic signature of cancer cells. The authors have done an admirable job in collating and presenting current knowledge in this field, which could be instrumental in developing new anticancer therapeutics. I have minor comments for your consideration.

I suggest dividing Figure 1 into two separate figures to enhance clarity or, alternatively, enlarging the font size within the figure for a clearer presentation.

Beyond glucose and amino acids, it would be beneficial to discuss other primary nutrients that may influence metabolic pathways. I recommend incorporating a section or a discussion on other potential components that could impact metabolic reprogramming in cancer cells.

It would be valuable to include an analysis of hypoxia and anaerobic glycolysis in cancer cells, as well as the presence of aerobic glycolysis in normal cells. This would provide a more comprehensive perspective on metabolic processes under different physiological conditions.

Could you elaborate on whether indoleamine 2,3-dioxygenase 1 (IDO1) plays a role in metabolic regulation within the context of your review? Considering the significance of IDO1 in tryptophan metabolism, its potential role in cancer cell metabolism and immune responses could enrich the discussion.

Author Response

This reviewer suggested increasing the font size of the letters in Fig. 1. We have done it now so that the letters are more legible.

Another suggestion was to include information on the role of other major nutrients in cancer. We already discussed glucose and amino acids in the original version. This leaves only fatty acids as the major nutrient. In response to the reviewer’s suggestion, we have added a paragraph at the end of this review under “Conclusions” on the role of fatty acids in cancer (page 20).

This reviewer also asked us to incorporate information on anerobic glycolysis in cancer cells and aerobic glycolysis in normal cells. In response to this comment, we revised the manuscript by adding a couple of paragraphs in the revised version (section 3.2.; page 6).

Another suggestion from this reviewer was to add some info about IDO1 and tryptophan metabolism. We have now revised the manuscript to address this comment by adding two paragraphs on the topic (section 3.9.5.; page 19). In this revised section, we also included some recent studies on methionine interaction between cancer cells and immune cells.

Round 2

Reviewer 3 Report

Comments and Suggestions for Authors

The authors have satisfactorily addressed the points which I noted.